# Involvement of Lipophagy and Chaperone-Mediated Autophagy in the Pathogenesis of Non-Alcoholic Fatty Liver Disease by Regulation of Lipid Droplets

**DOI:** 10.3390/ijms242115891

**Published:** 2023-11-02

**Authors:** Eleftheria M. Mastoridou, Anna C. Goussia, Panagiotis Kanavaros, Antonia V. Charchanti

**Affiliations:** 1Department of Anatomy-Histology-Embryology, Faculty of Medicine, School of Health Sciences, University of Ioannina, 45110 Ioannina, Greece; e.mastoridou@uoi.gr (E.M.M.); pkanavar@uoi.gr (P.K.); 2Department of Pathology, Faculty of Medicine, School of Health Sciences, University of Ioannina, 45110 Ioannina, Greece; agoussia@uoi.gr

**Keywords:** non-alcoholic fatty liver disease, lipophagy, chaperone-mediated autophagy, perilipins

## Abstract

Non-alcoholic fatty liver disease (NAFLD) is defined as the accumulation of lipids in the form of lipid droplets in more than 5% of hepatocytes. It is regarded as a range of diverse pathologies, including simple steatosis and steatohepatitis. The structural characteristics of lipid droplets, along with their protein composition, mainly including perilipins, have been implicated in the etiology of the disease. These proteins have garnered increasing attention as a pivotal regulator since their levels and distinct expression appear to be associated with the progression from simple steatosis to steatohepatitis. Perilipins are target proteins of chaperone-mediated autophagy, and their degradation is a prerequisite for lipolysis and lipophagy to access the lipid core. Both lipophagy and chaperone-mediated autophagy have significant implications on the development of the disease, as evidenced by their upregulation during the initial phases of simple steatosis and their subsequent downregulation once steatosis is established. On the contrary, during steatohepatitis, the process of chaperone-mediated autophagy is enhanced, although lipophagy remains suppressed. Evidently, the reduced levels of autophagic pathways observed in simple steatosis serve as a defensive mechanism against lipotoxicity. Conversely, in steatohepatitis, chaperone-mediated autophagy fails to compensate for the continuous generation of small lipid droplets and thus cannot protect hepatocytes from lipotoxicity.

## 1. Introduction

Non-alcoholic fatty liver disease (NAFLD) is the most common liver disorder worldwide, with a global prevalence of 25% [1]. NAFLD encompasses a spectrum of distinct pathologies characterized by the accumulation of predominantly triglycerides (TGs) within lipid droplets (LDs) in more than 5% of hepatocytes, with concurrent exclusion of alternate etiologies of hepatic lipid infiltration, including excessive alcohol consumption [2,3,4]. Non-alcoholic fatty liver (NAFL) refers to simple steatosis as the first, benign, and reversible stage, while 10–20% of cases progress to a more severe and inflammatory condition known as non-alcoholic steatohepatitis (NASH) [1]. Fibrosis may be present in around 30% of NASH cases and can evolve to an irreversible cirrhotic stage and finally to hepatocellular carcinoma (HCC) [5].

NAFLD strongly correlates with metabolic syndrome, as the pathogenesis of NAFLD is associated with various manifestations of the syndrome, such as insulin resistance, obesity, and dyslipidemia [6]. In accordance with this observation, in the early 2020s, the term “metabolic-dysfunction associated liver disease-MAFLD” was suggested to replace the term “NAFLD” [7]. Nevertheless, for the purpose of this manuscript, we will utilize the term “NAFLD”.

NAFLD not only has an increasing incidence worldwide, but also constitutes an important economic burden in the health system, as it is estimated to account for enormous medical costs in both European countries and the United States [1]. To date, there are no approved pharmacological treatments for NAFLD; thus, a better understanding of the pathogenetic mechanisms is more than urgent [8].

Various studies have elucidated the role of autophagy in the development of NAFLD [8,9,10]. Autophagy is a conserved catabolic process, which includes the degradation of dysfunctional proteins and organelles in the lysosomal lumen [11]. Autophagy consists of three main pathways: macroautophagy, microautophagy, and chaperone-mediated autophagy (CMA) [11]. Macroautophagy facilitates the engulfment of certain autophagic cargo, such as cytosolic organelles and proteins in autophagosomes, which will finally fuse with lysosomes and result in the degradation of the cargo. A selective form of macroautophagy, the so-called lipophagy, has been extensively studied, as a pivotal modulator in the progression of NAFLD [2]. Lipophagy involves the sequestration of LDs into autophagosomes, and their subsequent degradation is mediated by lysosomal lipases [12]. On the other hand, microautophagy and CMA do not involve autophagosomes. Microautophagy directly drives the fusion of the dysfunctional proteins into the lysosomes, while CMA is a more specific process, as it targets certain cytosolic proteins that contain the pentapeptide KFERQ motif to the lysosomes for degradation [13]. Macroautophagy and CMA are both constitutively active in cells, and they could be upregulated under stressful conditions [14].

Concerning the liver, both macroautophagy (hereafter referred to as “lipophagy”) and CMA have been associated with various physiologic and pathologic conditions; thus, these two processes will be the focus of this review. More specifically we will discuss the pathogenesis of NAFLD and provide an up-to-date review on the role of lipophagy and CMA in the development of the disease, mainly through the regulation of LDs.

## 2. Pathogenesis of NAFLD

The main hallmark of NAFLD is the accumulation of LDs in more than 5% of hepatocytes, which is known as steatosis [15]. NAFL is characterized by increased lipid accumulation, while NASH is characterized by steatosis, inflammation, hepatocyte injury, and possibly fibrosis [16].

Emerging evidence suggests that hepatic steatosis is a consequence of altered lipid metabolic processes due to increased accumulation of free fatty acids (FFAs) in the liver [17]. Insulin resistance seems to play a key pathophysiological role in the development of NAFLD. Under these conditions, insulin is no longer available to suppress lipolysis in adipose tissue; thus, increased circulation of FFAs results in enhanced efflux of lipids to the liver [2]. Esterification of FFAs into TGs, which are enclosed in LDs and stored in hepatocytes, constitutes an adaptive mechanism due to lipid overload through transformation of potentially toxic FFAs into neutral TGs [18]. Studies have shown that adipose tissue-derived fatty acids are the dominant source of hepatic TG accumulation in NAFLD [19,20]. In addition, as NAFLD has been associated with obesity and sedentary lifestyle, intestinal absorption of dietary lipids and subsequent accumulation in the liver in the form of LDs account for 16% of total lipid flux to the liver [20]. Another source of FFAs is de novo lipogenesis (DNL), a process through which the liver synthesizes fatty acids de novo, and these fatty acids are subsequently esterified into TGs. Although DNL is not abundant in the healthy liver, it contributes up to 26% of lipid accumulation in patients with NAFLD [20].

Until recently, the initial step in the progression of NAFLD has been considered the accumulation of fat in the liver (steatosis). This was proposed as the “two-hit hypothesis” by Day and James in 1998 [21]. Hepatic accumulation of lipids is considered as the first hit, which sensitizes the liver to further metabolic stressors. These metabolic stressors, such as mitochondrial dysfunction, endoplasmic reticulum (ER) stress, and inflammatory signals from circulating cytokines and adipokines, represent the second hit, which is a prerequisite for the progression to NASH [4].

However, this pathogenetic theory is currently considered outdated and too simplistic to delineate the development of NAFLD; thus, a more complex hypothesis has been widely accepted and termed the “multiple-hit hypothesis” [22]. According to this hypothesis, multiple parallel factors act synergistically in genetically predisposed individuals to promote NAFLD development [23]. This leads to mitochondria dysfunction with increased production of reactive oxygen species (ROS), the presence of elevated levels of ER stress, and activation of inflammatory cascades [24]. In addition, other metabolic factors, such as altered gut microbiota, which result in increased permeability and absorption of FFAs, contribute to enhanced circulation of toxic compounds and therefore to the release of proinflammatory cytokines [25]. Lipotoxicity, along with the activation of inflammatory pathways, leads to hepatocyte death, a crucial determinant that distinguishes NASH from NAFL [26]. Thereby, dead hepatocytes release factors that induce wound healing responses, such as the activation of resident immune cells (Kupffer cells) and hepatic stellate cells, leading intimately to the replacement of injured hepatocytes by fibrotic cells [27].

Both the two-hit and multiple-hit hypotheses consider NAFLD development as a progressive spectrum of pathological features, with steatosis being the primary and reversible stage, followed by steatohepatitis, a more severe inflammatory condition [22]. However, evidence suggest that NAFL and NASH may represent two independent conditions, without steatosis being a prerequisite stage for the progression to NASH [28]. In fact, some studies have documented that steatosis seems to occur in parallel with lipotoxicity; thus, NASH could potentially constitute the initial liver lesion, without an earlier stage of steatosis [4]. In the vast majority of patients with NAFLD, simple steatosis seems to be stable over time [29], with the histological progression to NASH being rare. Thus, some studies propose that NAFLD does not constitute a consecutive linear progression of different stages, but a complex metabolic liver disease with two distinct clinical entities, namely, NAFL and NASH [4].

Although steatosis has been considered as a totally benign situation, it seems that various disruptions that occur during increased accumulation of lipids in hepatocytes and contribute to the long-term development of lipotoxic conditions. More specifically, during steatosis, TGs constitute the main type of lipid enclosed in LDs. Studies have shown that TGs per se do not harm hepatocytes, although they are correlated with the severity of steatosis [30]. However, among TGs, other types of lipids, such as fatty acids, cholesterol, diacylglycerol, and phospholipids, accumulate in the liver and can injure hepatocytes [31]. The chronic accumulation of these lipids could promote lipotoxic conditions and therefore the progression of simple steatosis to steatohepatitis [32].

## 3. Role of LDs in NAFLD Progression

### 3.1. Biogenesis of LDs

In order to protect against the flux of elevated FFAs originating from the diet, adipose tissue and DNL to the liver, hepatocytes esterify FFAs in neutral lipids and store them in highly dynamic organelles, known as LDs [33]. Excess FFAs are primarily stored in the ER, where specific enzymes catalyze their esterification mainly into TGs, the most common neutral lipid found in hepatic steatosis, and cholesterol esters [17]. Once a critical concentration of neutral lipids is achieved, they begin to deform the ER bilayer, and they are finally stored in the cytoplasm [34]. Thus, LDs are composed of a hydrophobic core of neutral lipids, comprised predominantly of TGs and cholesterol esters, which are coated by a phospholipid monolayer along with specific proteins responsible for lipid metabolism [35].

### 3.2. Structural Changes in LDs Affecting NAFLD Progression

Structural characteristics of LDs, such as changes in lipid composition, have been associated with different stages of fatty liver disease. In fact, cholesterol esters in LDs have been found in liver biopsies of patients with NASH, but not in patients with NAFL, thereby pointing out a potential role of cholesterol esters in the promotion of inflammation and fibrosis [36].

Apart from lipid core composition, LD size also adapts to metabolic changes. Under normal conditions where the presence of LDs does not exceed 5% of hepatocytes, LDs appear small with a diameter of 100–200 nm [37]. However, histopathological studies from liver biopsies have shown that NAFL is commonly observed in the form of macrovesicular steatosis, which entails the accumulation of large droplets of fat within hepatocytes causing the nuclei to be displaced towards the periphery of the cytoplasm [38]. These large LDs are associated with higher quantities of TGs in the lipid core and morphological characteristics, such as low surface to volume ratio, which favors lipid storage rather than lipolysis and thus protects against the cytotoxic effects of FFAs [37,38]. On the other hand, studies from more severe steatohepatic stages have shown that NASH steatosis is usually seen as a coexistence of macrovesicular steatosis along with patches of microvesicular steatosis, referring to distended hepatocytes with a foamy, vacuolated cytoplasm, known as balloon cells [38]. In contrast to macrovesicular steatosis where the nucleus is displaced peripherally, the nucleus is usually centrally positioned in microvesicular steatosis [38]. Macrovesicular steatosis alone is considered a good long-term prognostic factor, as these patients rarely develop fibrosis or cirrhosis [38]. On the contrary, microvesicular steatosis has been found to co-exist with higher frequencies of severe macrovesicular steatosis, inflammation, cellular injury, and hepatocyte ballooning, features that are commonly found in NASH livers [39]. Thus, microvesicular steatosis could be considered as a significant prognostic factor of the transition from simple steatosis to steatohepatitis [40].

### 3.3. Perilipins: The Gatekeeper of LDs Surface and Their Role in NAFLD

The surface of LDs is coated by numerous proteins, such as adipose triglyceride lipase 1 (ATGL1), hormone-sensitive lipase (HSL) and perilipins (PLINs) [41]. PLINs, the best characterized and the most abundant LD-associated proteins, consist of five members: PLIN1 to PLIN5 [41]. PLIN1 is found primarily in mature adipocytes, PLIN2 and PLIN3 are found ubiquitously, PLIN4 is mainly restricted to adipocytes, while PLIN5 is found in oxidative tissues, such as those of the heart, muscle, and liver [33,41].

The expression of PLINs on the LD surface varies depending on the LD size, with small LDs expressing PLIN3, PLIN4, and PLIN5, medium LDs expressing PLIN2, and large LDs expressing PLIN1 [42]. PLIN1 is a key regulator of lipolysis and mediates exchange of lipids between LDs, thus contributing to the formation and stabilization of large LDs [43]. PLIN1 is exclusively expressed in the steatotic liver, while it is absent in normal healthy hepatocytes [44]. In fact, during adipocyte differentiation, the initially expressed PLIN2 in premature adipocytes is replaced by PLIN1 in mature adipocytes [41]. In line with this notion, PLIN1 is considered a marker of chronic steatosis, a condition characterized by LD maturation, and the sequential expression of PLIN3, PLIN5, PLIN2, and finally PLIN1 on the LD surface is noted during this gradual process [45]. In contrast to PLIN1, PLIN2 has been detected in few hepatocellular LDs of the normal liver, along with PLIN3 [44]. PLIN2 is the major hepatic LD protein as it is responsible for lipid accumulation [46] and plays a significant role in the accessibility of lipolytic mechanisms to LDs. More specifically, under energy deprivation, phosphorylation of PLIN2 in response to AMP-activated protein kinase (AMPK) activation acts as a recognition marker by heat shock cognate 71 kDa protein (HSC70), thereby contributing to the degradation of PLIN2 by CMA [47]. This modification allows cytosolic lipases, such as ATGL1 and autophagy-related proteins, to begin lipolytic processes [47]. In line with this notion, PLIN2 is the most upregulated PLIN in the fatty liver [44], and it has been proposed to be the most important marker of hepatic LD accumulation [48]. Straub et al. have not detected any significant correlation between the levels of PLIN1 and PLIN2 in steatosis compared to steatohepatitis [44]. PLIN3 is ubiquitously expressed and is mainly associated with LD biogenesis, while it has also antilipolytic properties and, similar to PLIN2, is degraded by CMA after its phosphorylation by AMPK [49]. Furthermore, PLIN5 is the most dynamic protein and is highly expressed in fasted hepatocytes [50]. During resting conditions or when cells are overloaded with lipids, PLIN5 acts as a gatekeeper since it inhibits ATGL-mediated lipolysis and mitochondrial beta-oxidation, serving as a protective factor against hepatic lipotoxicity [51]. On the contrary, during fasting conditions, PLIN5 upregulates lipolysis and mitochondrial beta-oxidation to cover energy demands [52]. Regarding NAFLD, studies have shown significantly increased levels of PLIN5 in severely steatotic liver [53], as it has been shown to be required for the adaptation to lipid overload and it has been proposed as a crucial regulator of LD metabolism, hepatic inflammation, and mitochondrial function [50]. Apart from simple steatosis, a limited number of studies have investigated the expression pattern of PLINs in NASH livers. Notably, researchers have pointed out that during steatohepatitis, PLIN2 exhibits major expression in small LDs, especially around ballooned hepatocytes, with the levels of expression being correlated with the severity of the inflammation [54]. Apart from PLIN2, levels of PLIN3 and PLIN5 on the surface of small LDs, as markers of acute hepatocellular injury, are also elevated, while PLIN1 is not been detected since its expression is observed in chronic steatotic conditions (Figure 1) [45].

### 3.4. Breakdown of Lipid Droplets

LDs undergo a biogenesis and degradation cycle that contributes to LD homeostasis. Dysregulation of LD homeostasis may result in increased intracellular lipid accumulation and thus the development of NAFLD [55]. Catabolism of LDs into FFAs and glycerol is induced under energy deprivation conditions; thus, LDs constitute a significant energy storage pool. Consequently, the tight regulation of LD metabolism is crucial, especially for energy-dependent organs, such as the liver [35]. Lipolytic pathways include lipolysis, which is mediated through cytosolic lipases, such as ATGL, and lipophagy, a selective breakdown process of LDs by macroautophagy components [56].

#### 3.4.1. Degradation of PLINs via CMA

LD degradation is mediated by two major lipolytic pathways: lipolysis by neutral lipases and lipophagy by acid lysosomal lipases [57]. As mentioned above, numerous proteins on the LD surface play major roles in LD homeostasis and communicate with other organelles [58]. Along with this notion, studies have focused on PLINs and their role in LD degradation, with PLIN2, PLIN3, and PLIN5 acting as negative regulators of lipolytic mechanisms; therefore, the dissociation of these proteins from the LD membrane is a prerequisite for the accessibility of lipolytic mechanisms [59,60]. Further studies have shown the presence of the characteristic KFERQ motif in PLIN2, PLIN3, and PLIN5, which have been identified as CMA substrates [53,61]. CMA constitutes a selective form of autophagy by specifically targeting proteins containing the KFERQ motif and delivering this cargo directly to the lysosomes based on the interaction of HSC70 with lysosomal-associated membrane protein 2A (LAMP2A). HSC70 recognizes the characteristic motif in the PLINs, and together they interact with the cytosolic tail of LAMP2A, thus mediating the translocation of PLINs into the lysosomal lumen [62]. CMA constitutes a significant metabolic regulator in the liver as it is activated under various stressful conditions, such as starvation or in response to lipid overload [63]. However, during chronic lipid dysregulation, as observed in NAFL, CMA is downregulated, mostly due to changes in the lipid composition of the lysosomal membrane [64].

#### 3.4.2. Degradation of the Lipid Core via Lipolysis and Lipophagy

After the removal of PLINs, LDs are accessible to lipolytic mechanisms. Lipolysis occurs in the cytoplasm by neutral lipases that directly act on the lipid core of LDs to produce FFAs, a substrate for mitochondrial beta-oxidation [65].

On the other hand, lipophagy occurs inside the lysosomes and involves the autophagic sequestration of LDs in the autophagosomes and the subsequent fusion with lysosomes [57]. LDs are finally degraded by lysosomal lipases, and FFAs are released for energy production. Lipolysis and lipophagy seem to be concomitantly activated; however, studies have underlined a potential preference for lipolysis in larger LDs, whereas lipophagy targets smaller LDs [66]. Along this line, ATGL-mediated lipolysis has been proposed to act as an upstream pathway in larger LDs in order to produce small newly formed LDs by re-esterification that could be finally targeted by lipophagy [66]. Studies have shown that cytosolic lipases, such as ATGL, interact with key autophagic proteins, such as light chain (LC3B) and p62, to recruit LDs to autophagosomes [67]. Thus, it has been proposed that these two pathways not only work in tandem, but they are also cross-regulated. Liver-specific ATGL overexpression resulted in an increased abundance of lysosomal-associated proteins, such as LC3B, and reduced p62 levels, thus indicating that ATGL is not only a cytoplasmic lipase, but also a sufficient driver of lipophagy [68,69]. PLINs also play crucial roles in the regulation of lipophagy. As mentioned above, PLINs, especially PLIN2, act as inhibitors of lipophagy. Along with this notion, studies have shown that mice deficient in PLIN2 had increased levels of lipophagy; thus, the degradation of PLIN2 by CMA enhances lipophagy [70]. Apart from PLINs, several other LD-associated proteins have been proposed to regulate lipophagy, such as the family of small regulatory Rab GTPase (Rab) molecular switches. More specifically, under nutrient deprivation conditions, Rab7 has been postulated to enhance lipophagy by promoting the recruitment of lysosomes to the surface of LDs [71].

## 4. Autophagy Modulation during NAFLD

Steatosis, as mentioned above, is a benign and reversible stage of NAFLD, in which toxic FFAs are transformed into neutral TGs and finally stored as LDs in the cytoplasm of hepatocytes [18]. During the past decade, autophagy has gained significant attention, as a potential crucial regulator of the pathogenesis of NAFLD; thus, autophagy activators and inhibitors could possibly serve as promising therapeutic targets. It is well established that autophagy not only regulates the degradation of damaged organelles and cytosolic components, but also various metabolic processes. For instance, lipid balance, is maintained mainly through lipophagy, while CMA has emerged as a potential co-regulator. Both lipophagy and CMA are constitutively active in hepatocytes, thereby contributing to lipid homeostasis [14].

### 4.1. Lipophagy Modulation during NAFLD

Various studies have suggested the importance of lipophagy in the maintenance of liver homeostasis, as it mediates the catabolism of LDs to FFAs [17]. Lipophagy plays a crucial role not only in the regulation of lipid transportation, but also in the adaptation of cells to several insults, such as lipid imbalance [9]. It is well established that lipophagy activity is impaired in NAFLD [72].

Immunohistochemical studies have demonstrated inhibition of lipophagy in human liver samples diagnosed with NAFLD [73]. In accordance with this observation, increased levels of LC3-II and p62, the most highly studied lipophagy markers, have been found in patients with NAFLD [74]. Under conditions of energy surplus and a high-fat diet, mammalian target of rapamycin (mTOR), a widely known negative regulator of lipophagy, is frequently hyperactivated, thus inhibiting initiation of lipophagy [75]. However, the long-term inhibition of lipophagy in the steatotic liver has been found to be mediated by the downregulation of autophagy-related transcription factors, such as forkhead box transcription factors (FOXO) and transcription factor EB (TFEB) [76,77]. Consequently, the steatotic liver is characterized by decreased expression of genes associated with an autophagic core mechanism and the formation of autophagosomes [9]. Additionally, a disturbance in lysosome biogenesis and an impediment in autophagosome-lysosome fusion due to alterations in the cellular membrane lipid composition of autophagosomes and lysosomes have been reported [78].

Despite the well-established notion that lipophagy is impaired in NAFLD, there is limited evidence on the levels of lipophagy through the different stages of NAFLD. A recent study has demonstrated that lipophagy is impaired in the later stages of NAFLD in both in vivo and in in vitro models [8]. Further investigations of LC3-II and p62 protein levels have shown increased lipophagy flux in the early stages of NAFL with a gradual decrease during disease progression [79]. Similar findings have been reported in human liver specimens suffering from NASH, as evidenced by elevated levels of LC3-II and p62, thus indicating suppression of lipophagy, with the perpetual accumulation of LC3-II and p62 being positively correlated with the severity of the disease [74].

### 4.2. CMA Is Involved in LD Degradation during NAFLD

Recently, the role of CMA has emerged concerning the metabolic functions of the liver [80]. CMA has been shown to be associated with lipid homeostasis, while mice with defects in liver CMA have been shown to develop hepatosteatosis [10]. CMA substrates involve only proteins, not lipids; however, it is strongly associated with LD metabolism. Intriguingly, PLIN2, PLIN3, and PLIN5 have been found to contain the pentapeptide KFERQ [53]. Therefore, they interact with HSC70, promoting the delivery of PLINs to lysosomes for selective degradation [61]. In fact, mutation of the KFERQ peptide in PLIN2 resulted in LD accumulation, thereby elucidating a crucial role for CMA as an upstream regulator of both lipolysis and lipophagy [59].

A limited number of studies have identified the role of CMA as a crucial regulator of hepatic metabolism and hepatic adaptation to stressful stimuli, such as energy deprivation and lipid overload [63]. The cellular response to various stressors, including prolonged starvation, oxidative stress, and exposure to factors that result in protein damage, conditions that characterize NASH livers, elicits maximal activation of CMA [64]. Constitutive blockade of CMA has been shown to induce hepatic steatosis, although the lipophagy machinery remained intact [63]. LAMP2A has been widely used as a marker of CMA activity. LAMP2A levels are elevated in mildly steatotic liver compared to normal liver, while these levels were significantly decreased in progressive steatosis, suggesting that CMA activity is inversely correlated with the severity of hepatosteatosis [53]. The initial elevated levels of CMA in cases of mild steatosis acted as a compensatory mechanism for lipid overload; however, when this lipid overload was prolonged, CMA was dramatically decreased [53]. A study by Rodriguez-Navaro et al. identified a negative impact of dietary lipid challenges on CMA activity. In fact, they suggested that this effect is primarily mediated by alterations in the lipid composition of the lysosomal membrane resulting from lipid exposure. Furthermore, they observed a decrease in the stability of lysosomal membrane proteins. Specifically, LAMP2A was found to be particularly susceptible to lipid composition changes, thereby revealing a distinct mechanism that contributes to compromised CMA under lipid overload conditions [64].

On the other hand, there is limited evidence on CMA levels in NASH disease. In fact, it is widely known that metabolic oxidative stress, primarily characterized by oxidatively modified proteins, triggers CMA, augments substrate translocation by HSC70 to the lysosomal membrane, and elevates LAMP2A levels [81]. Das et al. reported that LAMP2A levels were elevated in an induced mouse model of steatohepatic liver; therefore, this study proposes a pathogenetic model, wherein NASH is characterized by increased CMA levels [82]. That study reported that a subset of proinflammatory protein receptors, such as purinergic receptors, mainly the P2X7 receptor that undergoes upregulation in response to metabolic oxidative stress, may modulate the autophagy process by boosting, albeit to a limited extent, the mRNA levels of HSC70 and LAMP-2A and promoting the association of LAMP2A with the lysosomal membrane (Figure 2) [82]. However, the exact pathway that activates CMA in the NASH liver remains to be elucidated.

## 5. Discussion

NAFLD consists of distinct liver disorders that occur due to excessive accumulation of fat in the liver [23]. NAFLD manifests in various forms, ranging from simple fatty liver to NASH and the resulting fibrosis/cirrhosis [16]. Approximately one-fifth of NAFLD patients will experience disease progression to NASH, while roughly 20% will experience further progression to cirrhosis or to end-stage liver disease (ESLD), which can result in various complications and confer significant morbidity and mortality [83].

Recently, LDs have emerged as the focal point of studies pertaining to NAFLD pathogenesis. A range of LD characteristics, including their size, concentration of cholesterol esters, and expression of various PLINs in their lipid membranes, appear to contribute to liver steatosis [36,38]. It is noteworthy that different stages of NAFLD are distinguished by varying sizes of LDs and PLINs [42]. In the initial stages of NAFL, small LDs begin to accumulate in the cytoplasm and are primarily engulfed by PLIN2, a marker of liver steatosis [41]. However, in the later stages of NAFL, the size of LDs increases considerably, and PLIN2 is replaced by PLIN1, a hallmark of LD maturation and macrovesicular steatosis [45]. Furthermore, studies in NASH liver specimens have shown increased expression of PLIN2, PLIN3, and PLIN5, which are mainly found in small LDs [45,54]. In line with this notion, histopathological studies of NASH livers have pointed out the existence of a background of macrovesicular steatosis in conjunction with microvesicular steatosis patches, resulting in the foamy appearance of hepatocytes [45,54].

To date, various pathogenetic mechanisms have been identified as potential mechanisms that underlie NAFLD [16]. Autophagy pathways have been proposed as crucial regulators of hepatic steatosis and are implicated in lipid metabolism [9]. Therefore, various studies have pointed out the role of decreased levels of autophagy in the pathogenesis of NAFLD, but only a limited number of studies have investigated the levels of both lipophagy and CMA pathways during NAFLD progression from simple steatosis to NASH. In the initial stages of simple steatosis, there is an upregulation of lipophagy, which acts as a compensatory mechanism to counter the accumulation of lipids [8]. However, chronic lipid accumulation results in downregulation of lipophagy, which is mediated by different factors, such as mTOR activation and FOXO/TFEB downregulation [12,75]. Under these conditions, the accumulation of LDs is favored, but acts as a protective mechanism against lipotoxicity. Apart from NAFL, a limited number of studies have focused on the levels of lipophagy during NASH. Inflammation, oxidative stress, and lipotoxicity result in excessive ER stress that further downregulate lipophagy [74,84].

CMA has also been elucidated as a significant regulator of NAFLD pathogenesis [63]. The early stages of simple steatosis are characterized by increased levels of CMA, which acts in coordination with elevated lipophagy to compensate for lipid accumulation and restore hepatic homeostasis [53]. However, prolonged lipid overload results in downregulation of CMA by altering the lipid composition of the lysosomal membrane, thereby decreasing the stability of LAMP2A protein [64]. Therefore, we hypothesize that advanced stages of simple steatosis are characterized by decreased levels of both lipophagy and CMA, aiming to protect hepatocytes from oxidative stress and resulting from excessive FFAs production and subsequent lipotoxicity and ER stress.

On the other hand, during NASH development, inflammatory conditions are identified as excessive stressful stimuli that upregulate CMA, as evidenced by elevated levels of LAMP2A and HSC70 [82]. Oxidative stress, one of the key characteristics of NASH progression, activates CMA as a survival pathway in order to get rid of oxidized proteins and protect hepatocytes from apoptosis [85]. Although CMA seems to act as a compensatory mechanism for lipophagy inhibition in NASH liver, it is reasonable to hypothesize that the persistent biogenesis of small LDs coated by PLIN2, PLIN3, and PLIN5 may not be counteracted. Although these PLINs are substrates of CMA, it could be hypothesized that the levels of LD biogenesis exceed the capacity of CMA to degrade them. Therefore, the inhibition of lipophagy along with the continuous production of small LDs engulfed by PLIN2, PLIN3, and PLIN5 could be a proposed mechanism for NAFLD progression from simple steatosis to NASH. In line with this notion, a study by Asimakopoulou et al. demonstrated that tumor areas of HCC are characterized by markedly increased levels of PLIN5, thereby pointing out a potential role for the overexpression of this PLIN in disease progression [86].

Despite the evident progression on the understanding of the NAFLD pathogenesis from a molecular point of view, there is need to expand the research to more clinical aspects. Several studies have tried to elucidate potential regulators and risk factors of autophagic pathways in patients with NAFLD. Interestingly, hyperinsulinemia, found in patients with metabolic syndrome and insulin resistance, upregulates mTORC1, thus inhibiting the initiation of lipophagy and also suppressing the expression of various key autophagy genes [87]. However, whether insulin resistance is a cause or a consequence of NAFLD remains to be investigated [88].

Furthermore, interesting discoveries regarding autophagy signaling pathways in NAFLD may also have potential applications in the development of markers for detecting the progression of hepatic injury and in therapeutic approaches. Recent reports indicate that the measurement of serum p62/SQSTM1 levels has the potential to serve as a biomarker for the diagnosis of patients with steatosis and inflammation, indicative of NASH stage [89]. Within the context of NAFLD, the accumulation of lysosomes containing LDs, the final step in the lipophagy process, appears to be indicative of the gradual deterioration of lysosomal function, specifically the ability of lysosomal hydrolases to break down fat [88]. Taking all the above into consideration, the analysis of the compromised state of lipophagy using specific biomarkers could serve as a promising diagnostic tool in the assessment of patients with NAFLD [12]. Considering lipophagy as a potential therapeutic tool, various studies have tried to elucidate the impact of inducers of inhibitors in NAFLD progression (reviewed in [90]). Pharmacological activation of lipophagy at early stages has shown greater success in modifying the ultimate levels of steatosis [90]. On the contrary, when the manipulation of autophagic activity occurs at later stages, the outcomes become more susceptible to variations due to disparities in NAFLD progression [91]. However, there are several limitations regarding the use lipophagy as a therapeutic target in the clinical practice since supporting evidence concerning the relevance of lipophagy in human NAFLD is limited and primarily observational. Second, in order to minimize the likelihood of unintended autophagic side effects, it is necessary to identify specific and targetable regulators of lipophagy in the fatty liver without affecting the other tissues [2].

Hepatic lipophagy occurs in all hepatic cell types, including hepatocytes, macrophages, and hepatic stellate cells (HSCs). The role of HSCs has been extensively studied in liver fibrosis, wherein HSCs transdifferentiate from a quiescent state to myofibroblasts producing extracellular matrix and various profibrogenic factors, thereby promoting liver fibrosis [92]. Lipophagy plays also a crucial role in this procedure since HSCs in their quiescent state contain LDs in their cytoplasm [93]. The activation process begins with the autophagic degradation of these LDs through lipophagy [68]. Evidently, lipophagy has a detrimental effect on the liver given its effects on HSCs. On the contrary, induction of lipophagy in hepatocytes, especially in the early stages of NAFLD, protects against steatosis, thereby indicating a potential cell specific role of lipophagy in the liver. 

In addition to the liver, the role of lipophagy has been also investigated in the case of myocardial steatosis, where the role of PLINs also seems to be crucial. In contrast to the aforementioned role of PLINs in the regulation of lipophagy, controversial studies exist [94,95] as Ueno et al. pointed out that overexpression of PLIN2 induces myocardial steatosis [94], similar to its role in NAFLD. On the contrary, Mardani et al. demonstrated that PLIN2 deficiency results in reduced lipophagy, thus promoting the accumulation of LDs in the myocardium [95]. Therefore, since steatosis concerns multiple organs, the role of lipophagy and PLINs in different cells and tissues remains to be further clarified.

Although considerable advancements have been made in investigations of the pathogenesis of NAFLD, the evidence regarding the levels of lipophagy and mainly CMA during the progression of the disease from NAFL to NASH remains inadequate. Further research endeavors should concentrate on the precise signaling pathways that regulate both the levels of lipophagy and CMA and how these pathways are cross-regulated during the different stages of the disease. Furthermore, the histopathological characteristics of NAFLD livers should be further studied in correlation with the autophagic levels in order to elucidate the exact pathogenetic mechanism of macrovesicular and microvesicular steatosis. Ultimately, investigations of levels of PLINs along with other structural characteristics of LDs during the different stages of NAFLD could provide a more comprehensive understanding of this complex disease.

## Figures and Tables

**Figure 1 ijms-24-15891-f001:**
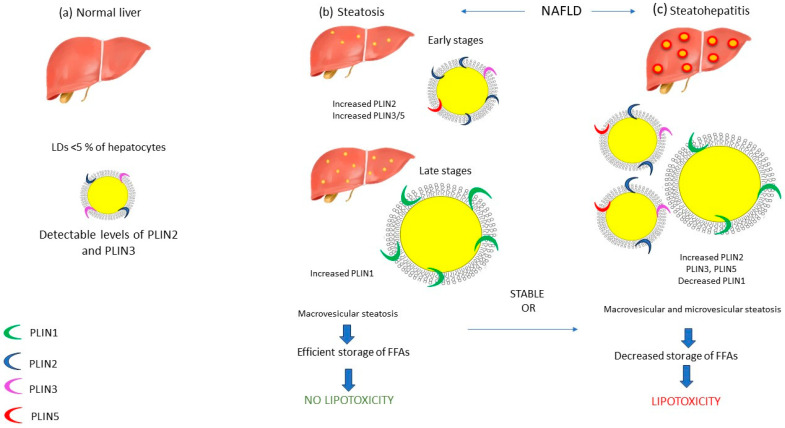
Perilipins and NAFLD. Under normal conditions, PLIN2 is detected in a few hepatocellular LDs along with PLIN3 (**a**). During the early stages of steatosis, small LDs begin to accumulate in hepatocytes coated by PLIN2, PLIN3, and PLIN5. Subsequently, under chronic lipid overload conditions, these PLINs are substituted by PLIN1, as LDs are becoming larger, a hallmark of LD maturation and macrovesicular steatosis (**b**). On the contrary, during the development of NASH, levels of PLIN2, PLIN3, and PLIN5 on the surface of small LDs are elevated and serve as markers of acute hepatocellular injury (**c**).

**Figure 2 ijms-24-15891-f002:**
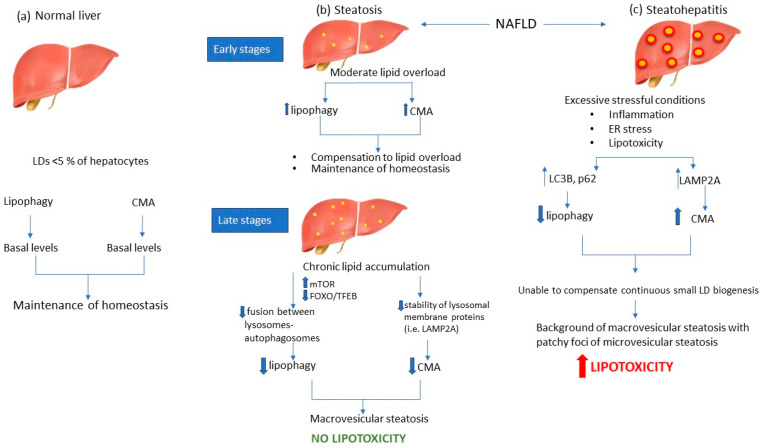
Autophagy modulation during NAFLD. In the normal liver, both lipophagy and CMA exist at basal levels to maintain lipid balance (**a**). During the early stages of steatosis, lipophagy and CMA are upregulated to compensate for increased lipid flux and therefore to inhibit LD accumulation and maintain lipid homeostasis. However, livers with advanced steatosis are characterized by decreased levels of lipophagy and CMA, mainly due to alterations in lysosomal lipid composition, resulting in reduced fusion between lysosomes and autophagosomes and reduced stability of lysosomal membrane proteins. Inhibition of autophagic mechanisms acts as a protective mechanism against lipotoxicity (**b**). On the other hand, NASH livers are distinguished by the presence of inflammation, oxidative stress, and lipotoxicity, which act as excessive stressful stimuli, thereby inhibiting lipophagy. At the same time, CMA is activated to alleviate the liver from the proteotoxic stress of oxidized proteins. However, the constant biogenesis of small lipid droplets cannot be counteracted by CMA levels, thereby leading to an uncontrollable lipotoxic environment (increased lipotoxicity) (**c**).

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
