# Peer review of "Involvement of Lipophagy and Chaperone-Mediated Autophagy in the Pathogenesis of Non-Alcoholic Fatty Liver Disease by Regulation of Lipid Droplets"

_ijms, 2023, doi:10.3390/ijms242115891_

Round 1
Reviewer 1 Report
Comments and Suggestions for Authors
NAFLD is still an emerging issue and its composed pathogenesis is still explored. We know about several hit theories and the development of oxidative stress, nevertheless lipotoxicity and chaperone-mediated autophagy have been raised recently as a novel causative factor . From a scientific and molecular point of view the manuscript appears to be well-written and based on appropriate literature. Nonetheless, I would like authors to expand the section of discussion with a context of possible clinical utility of the biochemical mechanisms in the course of NAFLD - especially according to the possibilities of diagnostics, monitoring and treatment with a special attention to lipophagy. Were there any attempts done in this area? Was lipophagy already treated as the target for treatment in the course of other pathologies?
Author Response
Reviewer 1
We would like to thank you very much for taking the time to review our manuscript. We took into consideration your comments and we have provided information concerning the potential clinical applications of lipophagy (Lines 451-472). In fact, in Lines 451-461 we have highlighted some important advancements in the application of lipophagy as a diagnostic tool, since several autophagic markers, such as levels of expression of p62/SQSTM1 and levels of lysosomes containing LDs have been correlated with the severity of NAFLD. Considering the role of lipophagy as a potential therapeutic target we have underlined some important information (Lines 461-472), such as the amelioration of steatosis with the usage of lipophagy inducers, especially during the early stages of NAFLD. However, these encouraging results are not supported when induction of lipophagy is applied in later stages of the disease. Finally, considering your comment on the role of lipophagy in other pathologies, we have provided some important information (Lines 484-491) about myocardial steatosis, since during metabolic syndrome apart from the liver, the heart is also affected. The study of lipophagy in myocardial steatosis has lead to controversial results, since PLINs, and especially PLIN2 which is the major PLIN expressed in the heart, has not the same role as in the liver. Studies have shown that PLIN2 could either induce or downregulate lipophagy in the heart thus preventing or promoting myocardial steatosis. The regulation of lipophagy seems to be tissue specific and remains to be further clarified.

Reviewer 2 Report
Comments and Suggestions for Authors
It is very well written review on autophagy mediated lipid turnover in NAFLD.
I have following suggestion to improve the review further:
1.The role of hormones in regulating lipophagy/CMA in context to NAFLD may be discussed.
2. Mechanistic details of Lipophagy induction can be elaborated further.
3. Cell type specific effects of Lipophagy/CMA (hepatocytes vs HSCs) can be discussed.
Author Response
Reviewer 2
We would like to thank you very much for taking the time to review our manuscript. We took into consideration your very fruitful comments and we have made the necessary modifications.
Comment 1 “The role of hormones in regulating lipophagy/CMA in context to NAFLD”
Since NAFLD is associated with insulin resistance and metabolic syndrome, we added some research results (Lines 443-450) concerning the impact of insulin in lipophagy. Hyperinsulinaemia has been associated with downregulation of lipophagy, by activating mTORC1 and concomitantly by suppressing the expression of key autophagy genes. Although there is considerable information about the regulation of lipophagy in the context of insulin resistance, our research about hormonal regulation of CMA did not yield significant information.
Comment 2 “Mechanistic details of lipophagy induction can be elaborated further”
Considering the mechanistic induction of lipophagy, we have added some further information in the Lines 275-285. We have pointed out that several signaling pathways are implicated in the induction of lipophagy, such as cytosolic lipases, especially ATGL. Apart from its direct interaction with LC3, ATGL overexpression has been linked to increased levels of lipophagy. Furthermore, downregulation of PLIN2 has been shown to induce lipophagy, while on the contrary, accumulation of PLIN2 results in hepatic steatosis by inhibiting lipophagy. In addition, small GTPase Rab7 has been also postulated to induce lipophagy, by facilitating the recruitment of lysosomes to the LDs’ surface.
Comment 3 “ Cell type specific effects of lipophagy/CMA (hepatocytes vs HSCs)”
HSCs are widely known for their role in liver fibrosis (Lines 473-483). Under normal conditions HSCs have a quiescent phenotype, with LDs in their cytoplasm. During their transdifferentiation into myofibroblasts, LDs are selectively degraded through lipophagy, thereby mediating their activated phenotype and their capacity to produce extracellular matrix. Evidently, induction of lipophagy in HSCs seems to be necessary for their activation process, thus promoting liver fibrosis and acting as a detrimental factor. On the other hand, induction of lipophagy in hepatocytes, especially during the early stages of NAFLD, has a protective effect by ensuring lipid homeostasis and preventing the progression of the disease. Therefore, lipophagy seems to have cell specific effects by exhibiting different effects in different cell types. Unfortunately, our research in the recent bibliography did not yield significant insights into the role of CMA in HSCs, although is exists at basal levels.
